# Kinematic Properties of a Twisted Double Planetary Chaotic Mixer: A Three-Dimensional Numerical Investigation

**DOI:** 10.3390/mi13091545

**Published:** 2022-09-17

**Authors:** Telha Mostefa, Aissaoui Djamel Eddine, Naas Toufik Tayeb, Shakhawat Hossain, Arifur Rahman, Bachiri Mohamed, Kwang-Yong Kim

**Affiliations:** 1Department of Mechanical Engineering, Ziane Achour University of Djelfa, Djelfa 17000, Algeria; 2Gas Turbine Joint Research Team, University of Djelfa, Djelfa 17000, Algeria; 3Department of Industrial and Production Engineering, Jashore University of Science and Technology, Jashore 7408, Bangladesh; 4Bangabandhu Textile Engineering College, Kalihati, Tangail 1970, Bangladesh; 5Department of Matter Sciences, Laghouat University, Laghouat 3000, Algeria; 6Department of Mechanical Engineering, Inha University, 100Inha-Ro, Michuhol-Gu, Incheon 22212, Korea

**Keywords:** twisted double planetary, active mixer, chaotic mixing, Poincaré section, kinematic properties, unsteady flow, high viscous fluid

## Abstract

In this study, a numerical investigation based on the CFD method is carried out to study the unsteady laminar flow of Newtonian fluid with a high viscosity in a three-dimensional simulation of a twisted double planetary mixer, which is composed of two agitating rods inside a moving tank. The considered stirring protocol is a “Continuous sine squared motion” by using the dynamic mesh model and user-defined functions (UDFs)to define the velocity profiles. The chaotic advection is obtained in our active mixers by the temporal modulation of rotational velocities of the moving walls in order to enhance the mixing of the fluid for a low Reynolds number and a high Peclet number. For this goal, we applied the Poincaré section and Lyapunov exponent as reliable mathematic tools for checking mixing quality by tracking a number of massless particles inside the fluid domain. Additionally, we investigated the development of fluid kinematics proprieties, such as vorticity, helicity, strain rate and elongation rate, at various time periods in order to view the impact of temporal modulation on the flow properties. The results of the mentioned simulation showed that it is possible to obtain a chaotic advection after a relatively short time, which can deeply enhance mixing fluid efficiency.

## 1. Introduction

The homogeneity of mixtures by chaotic advection in active mixers has long been used in much different engineering fields, such as petroleum engineering, polymer production in chemistry, and pharmaceutical engineering [1]. The mixing mechanism can be achieved by hydrodynamic or thermal methods. In all cases, a good comprehension of the physical mechanisms is the sign to design an effective mixing protocol. The mixing mechanism commences with a “heterogeneous” state and ends with a “homogeneous” state. The degree of this final homogeneity is linked to the efficiency of the mixing mechanism. A state of homogeneous mixing can be observed as heterogeneous on a small scale [2].

The mixing mechanism consists of two phases. The first is called “agitation phase”, in which the different regions of the fluid are sheared, stretched, and folded, which can enhance the scale of the scalar gradient in the fluid. The second phase is the phase where thermal or molecular diffusion heads to smooth these gradients. Thus, the aim of mixing is often to eliminate condensation or temperature concentration; heat or mass transfer develops either by turbulent flow or by chaotic advection.

Telha et al. [3] studied the kinematic properties of a viscous chaotic flow in a stirred tank for both continuously and alternatively modulated protocols. The authors explored the impact of the temporal modulation on the enhancement of the kinematic process of the fluid flow in terms of vorticity, strain rate, and elongation rate, and it was found that the alternatively mixing process can show a chaotic behavior.

Lasbet et al. [4] investigated the impact of geometry on the performance of the kinematic parameters of the velocity profile in a laminar chaotic flow. The results showed that a minimal hydraulic diameter is the most appropriate for maximizing the kinematic parameters such as vorticity, strain rate, stretching, and folding.

Our previous work demonstrated that an active mixer, if the flow is generated by moving walls [3], offers better governing of the flow through the choice of periods, amplitudes, phases, and directions of relative wall movements. Understanding the advection of particles in an incompressible laminar fluid flow is of great importance for technological applications. The scattering of pollutants in the atmosphere, sedimentation, and mixing are just a few applications [5]. Chaotic advection is considered a useful solution to achieve a more efficient mixing in high viscosity flows. However, it is not necessary to be time-dependent in a three-dimensional flow to achieve chaotic behavior and smooth flow is sufficient, but to create the chaotic motion of fluid particles in two-dimensional flows, the flow must be time-dependent [6].

From the literature survey, it is confirmed that a system is chaotic if it satisfies one of the following conditions: the system is sensitive to initial conditions, the system creates a horse shoe-like transformation, and the system creates a Baker’s transformation or the topological characterization of the system [7].

It is possible to define a first criterion of efficiency to have a chaotic regime. The authors observed that one of the conditions necessary for this creation is the sensitivity to the initial conditions. This would result experimentally in a very fast (exponential) elongation of the tracer segment. In this case, the length of the segment varies as $e^{\lambda t}$.

A single particle traveling along a chaotic path can explore a entire chaotic region. In a mixing context, this property of chaotic flows ensures that every particle will eventually visit all areas of the chaotic region. The short-lived frequency with which particles visit a particular region of the flow depends on their initial spatial position. Spatial position determines the amount of stretching and reorientation that a small fluid element undergoes at that location. Eventually, the continuous process of stretching and reorientation leads to the distribution of the particles throughout the chaotic region [8].

The area visited by particles in a chaotic flow can be illustrated by the use of plots known as Poincaré sections, a very common tool in dynamical systems theory [9]. An analogous definition of Poincaré sections can be applied to time for dependent periodic systems. After certain particles are tagged in the stream, system snapshots are taken at periodic time intervals and overlap on a single plot [10,11,12,13,14,15,16,17].

Regions of regular motion (also called islands or isolated regions) appear either as empty regions (if no particles were initially placed in them) or as sets of closed curves. The boundaries between regular and chaotic regions are known as KAM surfaces [8,9,10,11,12,13,14,15,16,17,18,19,20] and appear as closed curves in Poincaré sections. The passive mixers are mixers that create a chaotic flow with a simple geometric disturbance and without energy input [21,22,23,24,25,26,27,28].

The effects of the thermal condition of non-Newtonian fluids’ rheology was investigated by Naas et al. for C-shape chaotic geometry [29,30] and multi-layer micromixers [31]. They selected a highly efficient heat mixing system and better thermodynamic performance, for a rheology index ranging from 1 to 0.49. The results showed that the decrease in the heat transfer flow of shear-thinning flows was higher than that of Newtonian fluids.

Active mixers require an external source of energy to cause chaos in the flow [32,33,34,35,36], whereas the passive one does not. It depends on the complexity of the mixer shape.

Niederkorn and Ottino [37] studied, numerically and experimentally, the improvement of mixing in the annular flow between two eccentric cylinders for non-Newtonian viscoelastic and shear-thinning fluids.

Leprevost et al. [38] introduced the area between two confocal ovals as a new geometry for mixing. They showed that, by selecting appropriate periodic velocities for the walls of the two ovals, the current lines of the Stokes flow become chaotic and the particles can be diffused through the domain with a series of periodic oscillations.

Galaktionov et al. [39] solved analytically the stationary 2D Stokes’s flow inside a rectangular cavity with a cylinder inside. They concluded that the resulting flow is chaotic due to the movement of the upper and lower walls of the rectangular cavity and the rotation of the cylinder and that the mixing in this model is greater and faster compared to the cylinder less cavity.

El Omari and Le Guer [2] studied the effect of chaotic mixing on heat transfer in a two-rotor mixer. They reported that discontinuous rotor speeds improve mixing and heat transfer.

Hosseinalipour et al. [40,41] examined the mixing of a non-Newtonian flow inside a continuous chaotic mixer that consists of an eccentric helical rotor inside a cylindrical stator. They concluded that the eccentricity of the helical rotor and stator leads to the formation of chaotic flow and, as a result, better mixing.

Msaad et al. [42] studied the effect of the number of rotating rods on mixing and heat transfer. Their numerical results revealed that more rotating rods lead to a more uniform mixture. They have shown that the non-continuous rotation of the walls is one of the essential factors to generate a chaotic mixture. The effect of chaotic flow on the mixing of highly viscous fluids was studied by Shirmo hammadi and Tohidi [43]. It has been shown that the poor mixture occurs when the rotors rotate at the same and constant speed. Therefore, the rotors are varied as a sinusoidal function of time. With the disturbance generated in the rotational speed of the rotors, the secondary flow leading to the trapping of the fluid particles is eliminated and the mixing is improved. Aref [44] studied advection in a two-dimensional Stokes flow slowly modulated in time. Even at a very low Reynolds number, the motion of chaotic particles in a “laminar” flow is easily achievable.

Our contribution is to make a 3D numerical investigation of the evolution of kinematic properties and checking the mixing process with using a Poincaré map tool and Lyapunov exponent for a double planetary chaotic mixer agitated by a new efficient mechanism, called “twisted double planetary”. We are motivated in this work by the numerical study conducted by Telha et al. [3] for stirred tank chaotic mixer.

## 2. Geometrical Description and Modeling

The sketch of the geometrical shape used in the study is presented in Figure 1, in which a double planetary active mixer composed of two agitating rods has the same dimensions positioned vertically inside a cylindrical tank. The tank rotates around its revolution axe (axe.1) and the rods rotate around two axes; the first one is perpendicular to the mid-distance between their centers (axe.2), and the second one is the revolution axe of the tank (axe.1). The studied flow is three-dimensional, and the geometrical parameter values are summarized in Table 1.

The tank and the rods rotate with a continuously modulated velocity (a sine square function) to create a chaotic flow. The dynamical parameters of the studied case are shown in Table 2.

To maintain our mixing studies, we chose a Newtonian fluid with a high viscosity and a high Prandtl number, which is glycerin. Its thermo-physical properties are listed in Table 3.

The maximum angular velocities are fixed to $\Omega_{1}=60\;\mathrm{rpm},{\;\Omega }_{2}=6\;\mathrm{rpm}$ for the rods and $\Omega_{3}=6\;\mathrm{rpm}$ for the tank. The tangential velocity, then, is the same as U=300 mm/s. Therefore, the Reynolds number can be evaluated as:(1)Re=ρ.U.2.R3−R1μ=21.29

Due to the fluid’s characteristics, low Reynolds number, high Prandtl number, and unsteady boundary condition, the fluid flow can be considered as laminar and unsteady and the fluid as Newtonian and incompressible.

The imposed boundary conditions for the mixing mechanism are shown in Table 2.

The mass conservation (2) and Navier–Stokes Equation (3) were numerically solved using Fluent ANSYS and are given by the following equations, respectively:(2)$$\vec{\nabla }\;.\vec{V}=0$$
(3)$$\rho \frac{D\;\vec{V}}{\mathrm{Dt}}=-\vec{\nabla }P+\mu \nabla^{2}\;\vec{V}$$

The coupled scheme was utilized to achieve the pressure–velocity coupling while in the spatial discretization. A second-order upwind scheme was adopted for momentum. During the numerical simulations, it was considered that the convergence is obtained when the residues are less than $10^{-6}$ for the conservation equations.

## 3. Mesh Study

Our active mixer domain shape varies with time due to the motion of the boundaries; therefore, a dynamic mesh was used to model the studied fluid flow. By using the user-defined functions, UDFs, as a C script, we characterized the motions of the two rods and the tank by compiling and building the C script into the CFD code. In order to optimize the numerical calculation, we performed a study of the independence of results, and the five different cases were investigated carefully. The number of nodes varies from 28,491 to 1,318,435.

To perform grid independence studies, the velocity magnitude profile was evaluated by increasing the mesh densities until the velocities are superposed.

Figure 2 represents the development of the velocity magnitude versus X-coordinates for various mesh densities, which indicates an obvious difference between 28,491, 101,800, 207,509, 731,394, and 1,318,435 curves, and also found a negligible difference between 731,394 and 1,318,435. In a conclusion, the 731,394 density is refined enough to find precise results with reduced computing time.

## 4. Results and Discussion

### 4.1. Flow Characteristics

For a better description of the flow, we illustrate, in the figures below, the velocity vectors in the three plans XY, XZ, and YZ at 18 s (see Figure 3), which can provide an idea about the direction of the fluid particles at this moment and about the hydrodynamic state in general. The secondary flows are also illustrated. When the fluid moves in the geometrical perturbation as the considered mixer, a secondary flow will be generated by the existence of a centrifugal force. The secondary flow is more important for the disturbance having a chaotic protocol. In order to evaluate the secondary flow, the vortex intensity is expressed by the following expression:(4)$$\Omega_{average}=\frac{1}{S}\int \left|\Omega_{Z}\right|dS$$
where *S* is the area of cross section and $\Omega_{Z}$ is the vorticity at this area. By reason of the secondary flow influence, the transversal motion of the particles rises and the axial dispersion drops, subsequently enhancing the mixing.

We observed that there are created vortices next to the agitating twisted rods depending on the position of rods at every period (see the spiral graph in b in Table 2). Note that the discontinuity at some moments in some graphs represents the position of the twisted planetary rods (see Figure 4).

### 4.2. Poincaré Map

A Poincaré map is a tool based on applying mathematical analysis for indicating the behavior of trajectories of several different particles starting from their initial positions and for exploring the dynamics of a system. The appropriate positions of particles initially chosen indicate the behavior of the total fluid domain. After each time period, the particles change their positions and, thus, the end of each time period; the positions of the new particles are recorded and saved on file, and then aggregated in one map.

From the Poincaré map, we can gather whether the behavior is “regular” or “chaotic”. A regular behavior creates isolated regions that appear as closed curves or as an empty region. A chaotic behavior presents when a single particle changes its position along a chaotic path, which ensures that each particle, finally, visits the entire domain of the chaotic region.

In our study, for creating Poincaré map, nine initial massless particles from the fluid domain were selected, as illustrated in Figure 5, with the coordinates shown in Table 4.

As shown in Figure 6, the Poincaré map was calculated and presented at different time periods T = 3 s, 6 s, 12 s, and 18 s in three different planes (XY plane, ZX plane, and ZY plane).

We observed, at the instant *t* = 3 s, that the fluid is spinning almost in constant trajectories and it has not blended perfectly. By increasing time periods, we found that the system becomes more and more chaotic from the second period, which can be explained by the appearance and scattering fluid particles everywhere. It is noticeable that, at about 18 s (6τ), all areas of mixer were visited, there are almost no regular zones and the whole system becomes chaotic, which shows the impact of the twisted double planetary movement of rods with the movement in the opposite direction of the tank on the mixing process.

### 4.3. Lyapunov Exponent 

Lyapunov exponent is a reliable mathematical tool that describes the degree of separation with the time of very close particles in order to indicate the quality of the mixture.

To assess the divergence degree of a fluid’s particles, it is assumed that a vector of a specific particle is placed in an arbitrary position in the flow domain and stretched with it. In this case, the size of the vector is expressed by the following expressions: (5)$$\frac{dl_{x}}{dt}=\frac{\partial u}{\partial x}l_{x}+\frac{\partial u}{\partial y}l_{y}+\frac{\partial u}{\partial z}l_{z}$$
(6)$$\frac{dl_{y}}{dt}=\frac{\partial v}{\partial x}l_{x}+\frac{\partial v}{\partial y}l_{y}+\frac{\partial v}{\partial z}l_{z}$$
(7)$$\frac{dl_{z}}{dt}=\frac{\partial w}{\partial x}l_{x}+\frac{\partial w}{\partial y}l_{y}+\frac{\partial w}{\partial z}l_{z}$$

The size of the vector is obtained by integrating Equations (5)–(7) over time. If the initial vector is considered to be a vector of length 𝑙_0_, the stretching rate and Lyapunov exponent are equivalent to:(8)$${str}_{t}=\frac{\left|l_{t}\right|}{\left|l_{0}\right|},\;\lambda =\frac{1}{t}\ln\left({str}_{t}\right)$$

The divergence of two different paths of two particles with very small initial distance (Table 5) can be measured by a number called the Lyapunov exponent, which is related to the degree of divergence of these two fluid particles.

The sign of chaotic behavior is the positivity of the Lyapunov exponent. To calculate the mixing performance, nine groups of points (pairs) were chosen at the instant (*t* = 0 s); see Figure 7.

The degree of divergence was calculated over time and illustrated in Figure 8, which presents the exponential evolution of the Lyapunov exponent for the selected groups: G0, G1, G2, G3, G4, G5, G6, G7, and G8.

The illustrations for all groups in Figure 8 show that, after six periods (6τ), the obtained Lyapunov exponents are positive and greater than one at least for the majority of the selected nine groups. This proves the efficiency of the mixing mechanism (the twisted double planetary movement of rods in addition to the opposite direction movement of the tank) to create the chaotic behavior.

### 4.4. Kinematic Properties

By using CFD, fluent, strain rate, vorticity, helicity, and elongation were calculated at several time periods (1τ, 2τ, 3τ, 4τ, 5τ, and 6τ) in order to control the flow behavior (see Figure 9).

a.**Strain rate (deformation)**: It represents the change in the strain of a specific material over time. It is expressed by the following expression [46]:


(9)
$$D={\left[2{\left(\frac{\partial u}{\partial x}\right)}^{2}+2{\left(\frac{\partial v}{\partial y}\right)}^{2}+2{\left(\frac{\partial w}{\partial z}\right)}^{2}+{\left(\frac{\partial u}{\partial y}+\frac{\partial v}{\partial x}\right)}^{2}+{\left(\frac{\partial u}{\partial z}+\frac{\partial w}{\partial x}\right)}^{2}+{\left(\frac{\partial v}{\partial z}+\frac{\partial w}{\partial y}\right)}^{2}\right]}^{\frac{1}{2}}$$


We see that all the regions of fluid domain are enhanced by increasing of this property. The great value of this property is always near the agitating planetary rods.

b.**Vorticity:** It is an indicator of fluid particle rotation. It is a vector that describes the measure of the spinning motion at any point in the fluid. We notice also in Figure 10 that the rotation is important in the area near the agitating rods [46].


(10)
$$\Omega =\frac{1}{2}{\left[{\left(\frac{\partial w}{\partial y}-\frac{\partial v}{\partial z}\right)}^{2}+{\left(\frac{\partial w}{\partial x}-\frac{\partial u}{\partial z}\right)}^{2}+{\left(\frac{\partial v}{\partial x}-\frac{\partial u}{\partial y}\right)}^{2}\right]}^{\frac{1}{2}}$$


c.**Helicity:** It is simply the projection of a spin vector in the orientation of its momentum vector. Helicity is positive if the particle spin vector points and the momentum vector take the same direction, and it will be negative if the point is in the opposite direction. We see in Figure 11 that this property can take both positive and negative values, as it spreads throughout the flow domain; note that this property takes large values near the agitating rods [46]. 


(11)
$$H=\frac{\vec{V}.\vec{\Omega }}{V\Omega }$$


d.**Elongation rate:** Itis defined as the compression or the extension of fluid particle, i.e., it can take either negative or positive values. The generalized elongation rate ε can be expressed by the following Equation (12), which is calculated from the analytical derivatives of the velocity components, as follows.


(12)
$$\varepsilon =\frac{u^{2}\left(\frac{\partial u}{\partial x}\right)+v^{2}\left(\frac{\partial v}{\partial y}\right)+w^{2}\left(\frac{\partial w}{\partial z}\right)+u.v\left(\frac{\partial u}{\partial y}+\frac{\partial v}{\partial x}\right)+u.w\left(\frac{\partial u}{\partial z}+\frac{\partial w}{\partial x}\right)+v.w\left(\frac{\partial v}{\partial z}+\frac{\partial w}{\partial y}\right)}{u^{2}+v^{2}+w^{2}}$$


e.**Instantaneous kinematic properties:** In order to assess the performance and the efficiency of our twisted double planetary mixer, a comparison of the instantaneous properties (strain rate, vorticity and absolute elongation) was conducted with the study already conducted by Telha et al. [3] using a stirred tank. The results shown in Figure 12 below confirm the clear superiority of the present mixer compared with the other mixer, especially with regard to the strain rate and the vorticity, which proves the efficiency of this mixer.

## 5. Conclusions

In this study, a numerical study of the mixing mechanism of an active continuous twisted double planetary mixer of a three-dimensional flow was performed by chaotic advection. This was conducted in order to explore the impact of the efficiency of the chosen mixing protocol (the twisted double planetary motion of rods with the opposite rotation of the tank) on the enhancement of mixing and the evolution of kinematic properties of the fluid at a low Reynolds number, Re = 21.29.

The Poincaré section and the Lyapunov exponent are two reliable mathematical tools that were used for describing the chaotic behavior of our active mixer. The kinematic properties that can be characterized are the strain rate (deformation), vorticity magnitude (rotation), helicity, and elongation rate.

The Poincaré maps showed that, at T = 18 s (6τ), there are almost no regular zones and the entire system becomes chaotic.

The Lyapunov exponent method indicated that, after (6τ) of calculation, the Lyapunov exponent tends to be positive and greater than one for the majority of the chosen nine groups of particles.

The kinematic properties of the studied flow are important and as it is known that the progress of the mixing level depends on the growth of kinematic properties. As a result, these properties showed a relative chaotic system of the fluid flow.

## Figures and Tables

**Figure 1 micromachines-13-01545-f001:**
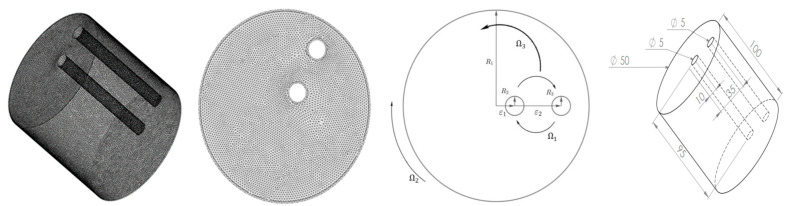
The geometrical shape of the considered double planetary mixer.

**Figure 2 micromachines-13-01545-f002:**
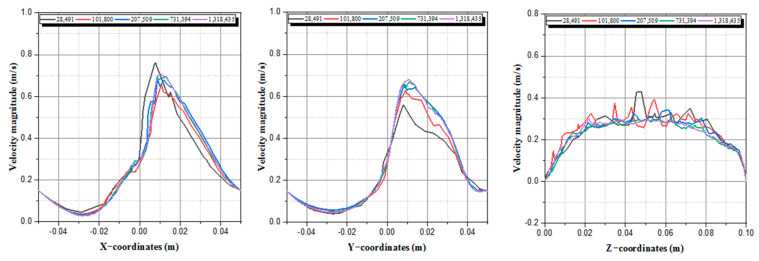
Velocity magnitude profiles versus X, Y, and Z coordinates for various mesh densities at t = 3 s.

**Figure 3 micromachines-13-01545-f003:**
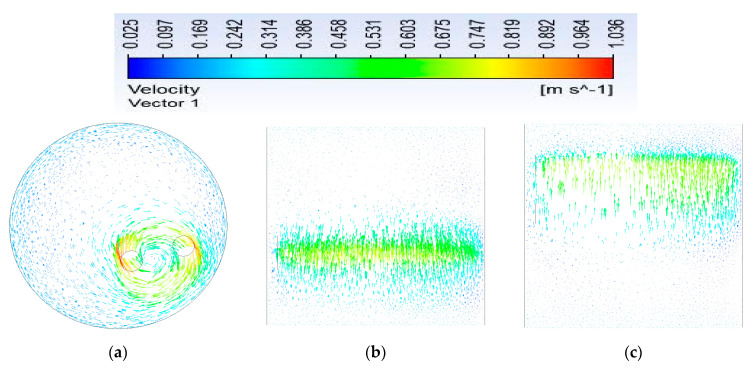
Velocity vectors in the (**a**) XY, (**b**) YZ, and (**c**) XZ planes at 18 s (6τ).

**Figure 4 micromachines-13-01545-f004:**
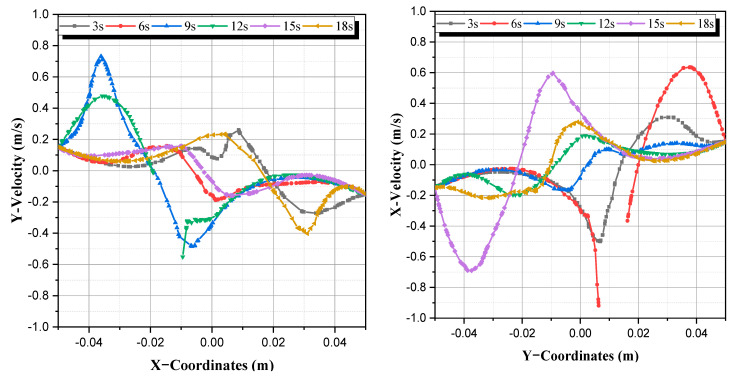
Profiles of the secondary flows at different periods.

**Figure 5 micromachines-13-01545-f005:**
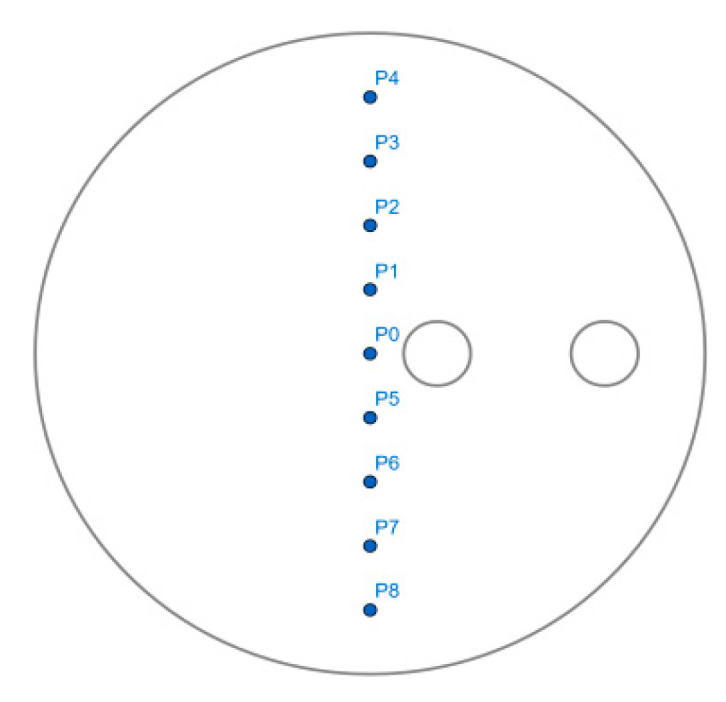
Initial selected positions of the Poincaré map.

**Figure 6 micromachines-13-01545-f006:**
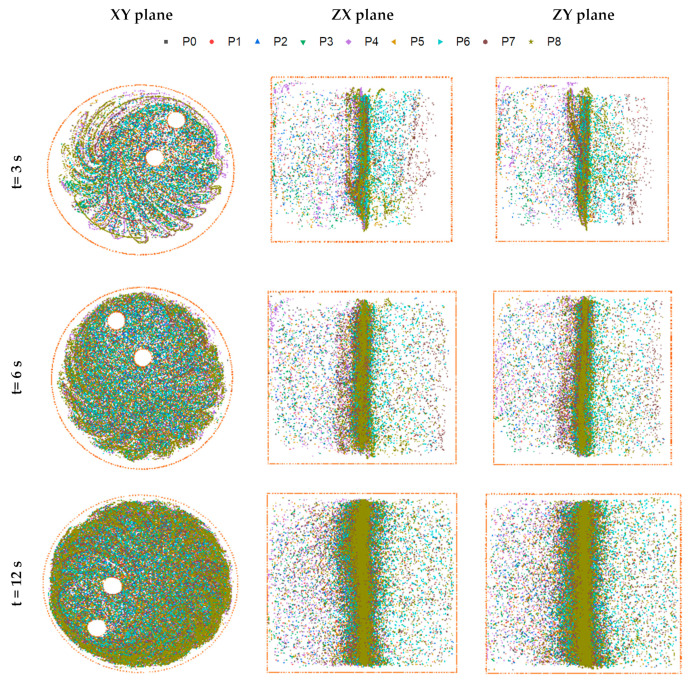
Poincaré map for the twisted double planetary mixer at different periods (3 s, 6 s, 12 s, and 18 s).

**Figure 7 micromachines-13-01545-f007:**
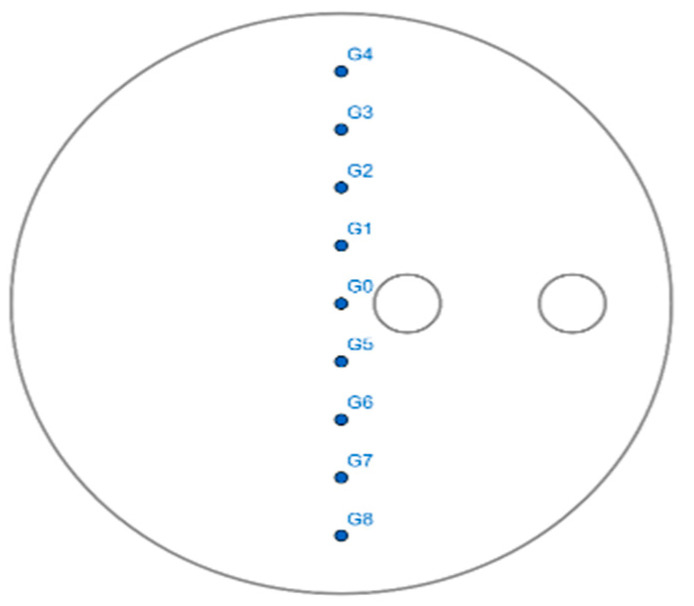
Initial chosen groups for the Lyapunov exponent method.

**Figure 8 micromachines-13-01545-f008:**
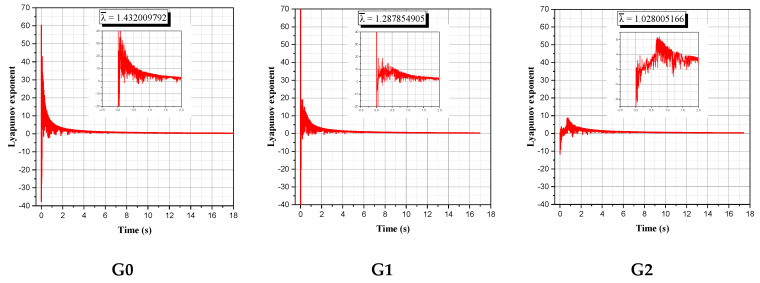
Evolution of the Lyapunov exponent for the twisted double planetary mixer after 18 s.

**Figure 9 micromachines-13-01545-f009:**
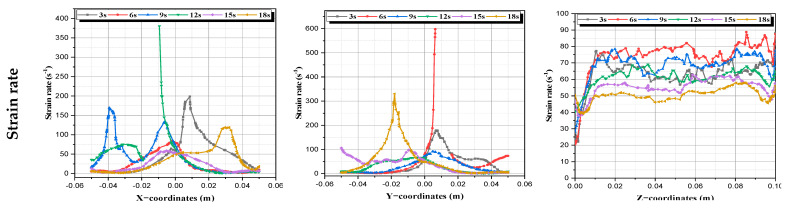
Strain rate profiles at every time period.

**Figure 10 micromachines-13-01545-f010:**
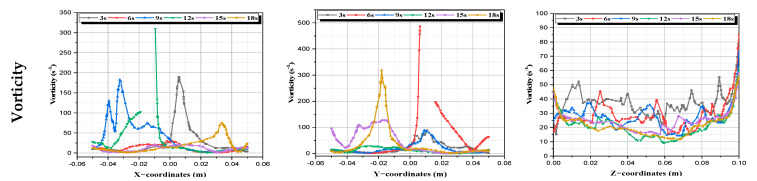
Vorticity profiles at every time period.

**Figure 11 micromachines-13-01545-f011:**
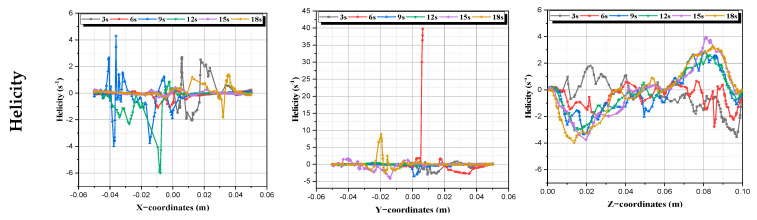
Helicity profiles at every time period.

**Figure 12 micromachines-13-01545-f012:**
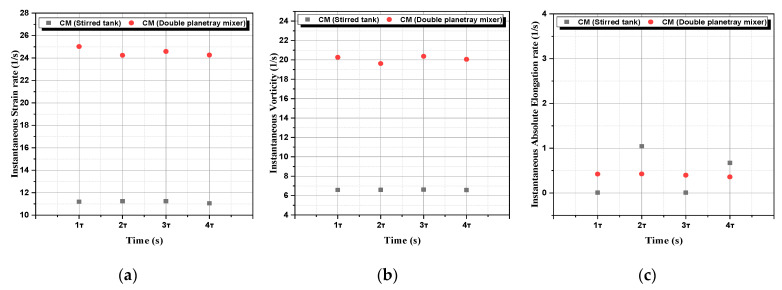
Instantaneous kinematic properties: (**a**) strain rate, (**b**) vorticity, and (**c**) absolute elongation.

**Table 1 micromachines-13-01545-t001:** Geometrical parameters.

$R_{1}$	Radius of Cylindrical Tank	50 mm	$\varepsilon_{2}$	Eccentricity of Rod 2	35 mm
$R_{2}$	Radius of cylindrical rod 1	5 mm	$H_{1}$	Hauteur of tank	100 mm
$R_{3}$	Radius of cylindrical rod 2	5 mm	$H_{2}$	Hauteur of rods	95 mm
$\varepsilon_{1}$	Eccentricity of rod 1	10 mm			

**Table 2 micromachines-13-01545-t002:** Dynamical parameters: (a) velocities of the active walls and (b) positions of rotating rods in a spiral graph at every time period.

Tank	Revolution Axe of Tank ($\Omega_{2}$)	$-\left(6-3\sin^{2}\left(\frac{\pi t}{\tau }+\frac{\pi }{2}\right)\right)$	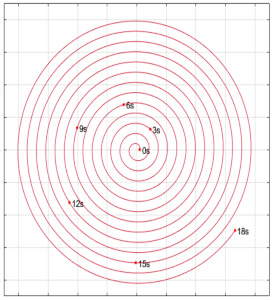
Rod 1	Axe 1 ($\Omega_{1}$)	$-\left(60-30\sin^{2}\left(\frac{\pi t}{\tau }\right)\right)$	
	Revolution axe of the tank ($\Omega_{3}$)	$6-3\sin^{2}\left(\frac{\pi t}{\tau }+\frac{\pi }{2}\right)$	
Rod 2	Axe 1 ($\Omega_{1}$)	$-\left(60-30\sin^{2}\left(\frac{\pi t}{\tau }\right)\right)$	
	Revolution axe of the tank($\Omega_{3}$)	$6-3\sin^{2}\left(\frac{\pi t}{\tau }+\frac{\pi }{2}\right)$	
	(a)		(b)

**Table 3 micromachines-13-01545-t003:** Glycerin’s thermo-physical properties [45].

**Dynamic Viscosity (** μ **)**	0.799 Pa.s	**Specific Heat (**$C_{p}$)	2427 J/Kg.K
Density (ρ)	$1259.9\;\mathrm{Kg}/m^{3}$	Thermal conductivity (k)	0.286 W/m.K
Prandtl number (Pr)	6780.325	Peclet number (Pe)	144,353

**Table 4 micromachines-13-01545-t004:** Initial coordinates for the Poincaré section.

**Particle**	**P0**	**P1**	**P2**	**P3**	**P4**	**P5**	**P6**	**P7**	**P8**
X	0	0	0	0	0	0	0	0	0
Y	0.04	0.03	0.02	0.01	0	−0.01	−0.02	−0.03	−0.04
Z	0.05	0.05	0.05	0.05	0.05	0.05	0.05	0.05	0.05

**Table 5 micromachines-13-01545-t005:** Initial coordinates of groups for the Lyapunov exponent.

	P1			P2		
	X	Y	Z	X	Y	Z
G0	0	0	0.05	0	0.001	0.05
G1	0	0.01	0.05	0	0.011	0.05
G2	0	0.02	0.05	0	0.021	0.05
G3	0	0.03	0.05	0	0.031	0.05
G4	0	0.04	0.05	0	0.041	0.05
G5	0	−0.01	0.05	0	−0.011	0.05
G6	0	−0.02	0.05	0	−0.021	0.05
G7	0	−0.03	0.05	0	−0.031	0.05
G8	0	−0.04	0.05	0	−0.041	0.05

## Data Availability

Data are available upon reader’s request.

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
