# Peer review of "Kinematic Properties of a Twisted Double Planetary Chaotic Mixer: A Three-Dimensional Numerical Investigation"

_micromachines, 2022, doi:10.3390/mi13091545_

Round 1

Reviewer 1 Report

The paper is about the CFD simulation of a double planetary chaotic mixer. The moving mesh was utilized to simulate high-viscous Newtonian liquid i.e. glycerin in the mixture. In this regard, the subject of the study is sufficiently new and interesting. The structure of the paper is good. and fluently presented. Few comments can be recommended to improve the paper quality:

1) it must be carefully spell-checked. Most words are connected to each other. I have tried to specify some of them in the attached file.

2) There are some paragraphs with only one sentence that is not correct.

3) Reference for table 3 shall be mentioned.

4)In Fig. 2: The time of the results should be mentioned.

5) In Fig. 3: a contour plot is redundant beside the vector plot and doesn't show new information. I recommend removing it. Also please provide a color legend for the vector plot.

6) Other comments can be found in the attached file.

Reviewer 2 Report

Brief summary:

The authors present a CFD simulation of a twisted double planetary mixer at a Reynolds number. The reviewer has some concerns about this study's intellectual merit and practical implications. The presentation of this paper should also be improved. Therefore, the reviewer does not recommend publication in its existing form.

General comments:

Many formatting issues in the manuscript require careful edits. What is the implication of this study? A CFD simulation is performed and presented with various quantities for characterizing the flow behavior. But these results are also only applied to the studied system. What generalizable information can be derived from this study?

Specific comments:

Fig. 1: an example of the computational mesh should also be presented.

Eq. 4: where is this equation used?

Line 186-190, the format and sentence structures need to be checked.

Line 187, the authors mention the volume fraction. Where is the equation for the volume of fraction? And why the volume of fraction is needed.

Line 187: what is the time step for the unsteady simulation? And how the time step is determined?

Line 200, this sentence is confusing. Please revise

Fig. 2: please report the numerical convergence study by following well-established guidelines. https://asmedigitalcollection.asme.org/fluidsengineering/article/130/7/078001/444689/Procedure-for-Estimation-and-Reporting-of

Fig. 3: how are the y-z plan and x-z defined? Also, why the velocity contour is not smooth?

Line 218: what is the definition of secondary flow? Are the authors referring to the Prandtl secondary flow? What is the cause for the secondary flow in this case?

Fig. 4: what is the elaboration on this figure?

Line 273: missing space

Line 276: missing space

Line 285: missing space

Round 2

Reviewer 1 Report

Unfortunately, the authors did not check carefully the manuscript for language and structure. There are many words that are connected to each other (at least in the PDF file) for example "confirmedthat" in line 77 or "theory[9].An" in line 99 and many others. you can find paragraphs with only one short sentence like line 107, line 201, or line 248. Although it can be recommended scientifically, I can not recommend it in its current format.

Reviewer 2 Report

The reviewer appreciates the author's efforts in the response. However, the reviewer does not think the comments have been sufficiently addressed. For example, the general comments have been completely ignored. The revised manuscript is also not track-changed. In addition, the fact that the authors randomly include the irrelevant volume of fraction in this study makes me wonder about this study's validity and reproducibility. Furthermore, some well-known terminality, such as Prandtl's secondary flow first-kind, seem to be unfamiliar to the authors.

Round 3

Reviewer 2 Report

Best wishes
